# Classification of Lidar Measurements Using Supervised and Unsupervised Machine Learning Methods

Ghazal Farhani[1,*], Robert J. Sica[1], and Mark Joseph Daley[2]

[1]Department of Physics and Astronomy, The University of Western Ontario, 1151 Richmond St., London, ON, N6A 3K7
[2]Department of Computer Science, The Vector Institute for Artificial Intelligence, The University of Western Ontario, 1151 Richmond St., London, ON, N6A 3K7

**Correspondence:** sica@uwo.ca

**Abstract.** While it is relatively straightforward to automate the processing of lidar signals, it is more difficult to choose periods of "good" measurements to process. Groups use various ad hoc procedures involving either very simple (e.g. signal-to-noise ratio) or more complex procedures (e.g. Wing et al. 2018) to perform a task that is easy to train humans to perform but is time consuming. Here, we use machine learning techniques to train the machine to sort the measurements before processing. The presented methods is generic and can be applied to most lidars. We test the techniques using measurements from the Purple Crow Lidar (PCL) system located in London, Canada. The PCL has over 200,000 raw profiles in Rayleigh and Raman channels available for classification. We classify raw (level-0) lidar measurements as "clear" sky profiles with strong lidar returns, "bad" profiles, and profiles which are significantly influenced by clouds or aerosol loads. We examined different supervised machine learning algorithms including the random forest, the support vector machine, and the gradient boosting trees, all of which can successfully classify profiles. The algorithms where trained using about 1500 profiles for each PCL channel, selected randomly from different nights of measurements in different years. The success rate of identification, for all the channels is above 95%. We also used the t-distributed Stochastic Embedding (t-SNE) method, which is an unsupervised algorithm, to cluster our lidar profiles. Because the t-SNE is a data driven method in which no labelling of training set is needed, it is an attractive algorithm to find anomalies in lidar profiles. The method has been tested on several nights of measurements from the PCL measurements. The t-SNE can successfully cluster the PCL data profiles into meaningful categories. To demonstrate the use of the technique, we have used the algorithm to identify stratospheric aerosol layers due to wildfires.

## 1 Introduction

Lidar (Light Detection and Ranging) is an active remote sensing method which uses a laser to generate photons that are transmitted to the atmosphere and are scattered back by atmospheric constituents. The back-scattered photons are collected using a telescope. Lidars provide both high temporal and resolution profiling, and are widely used in atmospheric research. The recorded back-scattered measurements (also known as level-0 profiles) are often co-added in time and/or in height. Before co-adding, profiles should be checked for quality purposes to remove "bad profiles". Bad profiles include measurements with low power laser, high background counts, outliers, and profiles with distorted or unusual shapes for a wide variety of instrumental or atmospheric reasons. Moreover, depending on the lidar system and the purpose of the measurements, profiles with traces of

clouds or aerosol might be classified separately. During a measurement, signal quality can change for different reasons including changes in sky background, the appearance of clouds, and laser power fluctuating. Hence, it is difficult to use traditional programming techniques to make a robust model that works under the wide range of real cases (even with multiple layers of exception handling).

In this article we propose both supervised and unsupervised machine learning approaches for level-0 lidar data classification and clustering. ML techniques hold great promise for application to the large data sets obtained by the current and future generation of high temporal-spatial resolution lidars. ML has been recently used to distinguish between aerosols and clouds for the Cloud-Aerosol Lidar and Infrared Pathfinder Satellite Observations (CALIPSO) level-2 measurements (Zeng et al., 2018). Furthermore, Nicolae et al. (2018) used a Neural Network algorithm to estimate the most probable aerosol types in a set of data

obtained from European Aerosol Research Lidar Network (EARLINET). Both Zeng et al. (2018) and Nicolae et al. (2018) concluded that their proposed ML algorithms can classify large sets of data and can successfully distinguish between different types of aerosols.

A common way of classifying profiles is to define a threshold for the signal-to-noise ratio at some altitude: any scan that does not meet the pre-defined threshold value is flagged bad. In this method, bad profiles may be incorrectly flagged as good, as

they might pass the threshold criteria, but have the wrong shape at other altitudes. Recently, Wing et al. (2018) suggested that a Mann-Whitney-Wilcoxon rank-sum metric could be used to identify bad profiles. In the Mann-Whitney-Wilcoxon test, the null hypothesis that the two populations are the same is tested against the alternate hypothesis that there is a significant difference between the two populations. The main advantage of this method is that it can be used when the data distribution does not follow a Gaussian distribution. However, Monte Carlo simulations have shown that when the two populations have similar

medians, but different variances, the Mann-Whitney-Wilcoxon can wrongfully accept the alternative hypothesis (Robert and Casella, 2004). Here, we propose a machine learning (ML) approach for level-0 data classification. The classification of lidar profiles is based on supervised ML techniques which will be discussed in detail in Section 2.

Using an unsupervised ML approach, we also have examined the capability of ML to detect anomalies (traces of wildfire smoke in lower stratosphere). The PCL is capable of detecting the smoke injected to lower stratosphere from wildfires (Doucet,

2009; Fromm et al., 2010). We are interested to investigate if after major wildfires the PCL can automatically (by using ML methods) detect aerosol loads in the upper troposphere and lower stratosphere (UTLS). Aerosols in the UTLS and stratosphere have important impact in the radiative budget of the atmosphere. Recently, Christian et al. (2019) proposed that smoke aerosols from the forest fires, unlike the aerosols from the volcanic eruptions, can have a net positive radiative forcing. Considering that the number of occurring forest fires have increased, detecting the aerosol loads from fires in the UTLS and accounting for them

in atmospheric and climate models is important.

Section 2 is a brief description of the characteristics of the lidars we used and an explanation on how ML can be useful for the lidar data classification. Furthermore, the algorithms which are used in the paper are explained in detail. In Section 3 we show classification and clustering results for the PCL system. In Section 4, a summary of the ML approach is provided, and the future directions are discussed.

## 2 Machine Learning Algorithms

### 2.1 Instrument Description and Machine Learning classification of data

The PCL is a Rayleigh-Raman lidar which has been operational since 1992. Details about PCL instrumentation can be found in Sica et al. (1995). From 1992 to 2010, the lidar was located at the Delaware Observatory (42.5°N, 81.2°W) near London,

Ontario, Canada. In 2012, the lidar was moved to the Environmental Science Western Field Station (43.1°N, 81.3°W). The PCL uses a second harmonic of an Nd:YAG solid state laser. The laser operates at 532 nm and has a repetition rate of 30 Hz at 1000 mJ. The receiver is a liquid mercury mirror with the diameter of 2.6 m. The PCL currently has four detection channels:

1. A High Gain Rayleigh (HR) channel that detects the back scattered counts from 25 to 110 km altitude (vertical resolution 7 m).

2. A Low Gain Rayleigh (LR) channel that detects the back scattered counts from 25 to 110 km altitude. This channel is optimized to detect counts at lower altitudes where the high intensity back scattered counts can saturate the detector, and cause non linearity in the observed signal. Thus, using the low gain channel, at lower altitudes, the signal remains linear (vertical resolution 7 m).

3. A Nitrogen Raman channel that detects the vibrational Raman shifted back-scattered counts above 0.5 km in altitude

(vertical resolution 7 m).

4. A Water Vapour Raman channel that detects the vibrational Raman shifted back-scattered counts above 0.5 km in altitude (vertical resolution 24 m).

The Rayleigh channels are used for atmospheric temperature retrievals, and the water vapour and nitrogen channels are used to retrieve water vapour mixing ratio. .

In our lidar scan classification using supervised learning, we have a training set in which, for each scan, counts at each altitude are considered as an attribute, and the classification of the scan is the output value. Formally, we are trying to learn a prediction function $f(x) : x \rightarrow y$ which minimize the expectation of some loss function $L(y, f) = \Sigma_i^N (y_i^{true} - y_i^{predicted})$, where $y_i^{true}$ is the actual value (label) of the classification for each data point, and $y_i^{predicted}$ is the prediction generated from the prediction function and $N$ is the length of data-set (Bishop, 2006). Thus, the training set is a matrix with size (m,n), and

each row of the matrix presents a lidar scan in the training set. The columns of this matrix (except the last column), are photon counts at each altitude. The last column of the matrix shows the classifications of each scan. Examples of each scan's class for PCL measurements using Rayleigh and nitrogen Raman digital channels are shown in Fig. 1.

We also have examined unsupervised learning to generate meaningful clusters. We are interested in determining if the lidar profiles, based on their similarities (similar features), will be clustered together. For our clustering task, a good ML method will

distinguish between high background counts, low laser power profiles, clouds, and high laser profiles, and put each of these in a different cluster. Moreover, using unsupervised learning, anomalies in profiles (aka: traces of smoke in higher altitudes) should be apparent.

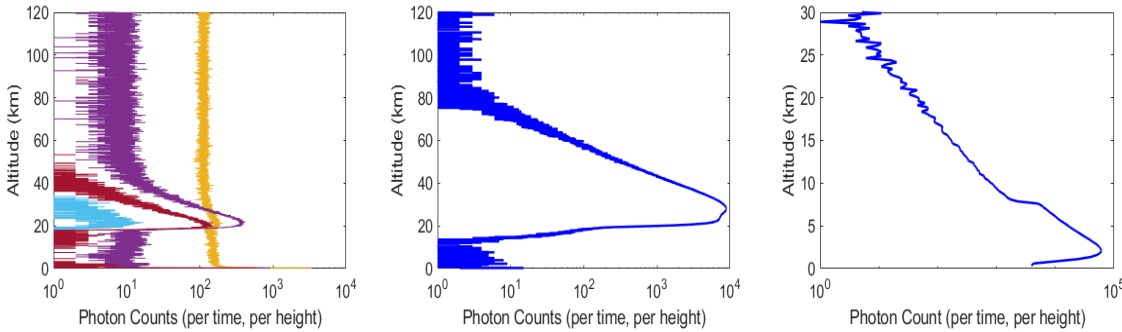

**Figure 1.** Example of measurements taken by PCL Rayleigh and Raman channels. Left panel: examples of bad profiles for both Rayleigh and Raman channels. In this plot, the signal in cyan and dark red have extremely low laser power, the purple signal has extremely high background counts, and the signal in orange has a distorted shape, and high background counts. Middle panel: example of a good scan in the Rayleigh channel. Right Panel: example of cloudy sky in the nitrogen (Raman) channel. At about 8 km a layer of either cloud or aerosol occurs.

Many algorithms have been developed for both supervised and unsupervised learning. In the following section, we introduce Support Vector Machine (SVM), Decision Trees, Random Forests, and Gradient Boosting Tree Methods as part of ML algorithms that we have tested for sorting lidar profiles. We also describe the t-distributed Stochastic Neighbour Embedding Method and Density-based spatial clustering as unsupervised algorithms which were used in this study.

Recently, Deep Neural Networks (DNNs) have received attention in the scientific community. In the Neural Network approach the loss function computes the error between the output scores and target values. The internal parameters (weights) in the algorithm are modified such that the error becomes smaller. The process of tuning the weights continues until the error is not decreasing anymore. A typical deep learning algorithm can have hundreds of millions of weights, input and target values. Thus, the algorithm is useful when dealing with large sets of images and text data. Although, DNNs are power full tools, they are acting as black boxes and important questions such as what features in the input data are more important remain unknown. For this study we decided to use the classical machine learning algorithms as they can provide better explanation of feature selection.

## 2.2 Support Vector Machine Algorithms

SVM algorithms are popular in the remote sensing community because they can be trained with relatively small data sets, while producing highly accurate predictions (Mantero et al., 2005; Foody and Mathur, 2004). Moreover, unlike some statistical methods such as the Maximum Likelihood Estimation that assume the data is normally distributed, SVM algorithms do not require this assumption. This property makes them suitable for data sets with unknown distributions. Here, we briefly describe how SVM works. More details on the topic can be found in Burges (1998) and Vapnik (2013).

The SVM algorithm finds an optimal hyperplane that separates the data set into a distinct predefined number of classes (Bishop, 2006). For binary classification in a linearly separable data-set, a target class $y_i \in \{1, -1\}$ is considered with a set of

input data vectors $x_i$. The optimal solution is obtained by maximizing the margin ($w$) between the separating hyperplane and the data. It can be shown that the optimal hyperplane is the solution of the constrained quadratic equation:

$$minimize: \frac{1}{2}||w||^2 \tag{1}$$

$$constraint: y_i(w^\intercal x_i + b) \geqslant 1. \tag{2}$$

5    In the above equation the constraint is a linear model where $w$ and the intercept (b) are unknowns (need to be optimized). To solve this constrained optimization problem, the Lagrange function can be built:

$$L(w, b, \alpha) = \frac{1}{2}\left\|w^2\right\| - \sum_i \alpha_i \left(y_i(w^\intercal x_i + b) - 1\right) \tag{3}$$

where $\alpha_i$ are Lagrangian multipliers. Setting the derivatives of $L(w, b, \alpha)$ with respect to $w$ and b to zero:

$$w = \sum_i \alpha_i y_i x_i \tag{4}$$

$$\sum_i \alpha_i y_i = 0. \tag{5}$$

Thus we can rewrite the Lagrangian as:

$$L(w, b, \alpha) = \sum_i \alpha_i - \frac{1}{2}\sum_i \sum_j \alpha_i \alpha_j y_i y_j x_i^\intercal x_j \tag{6}$$

It is clear that the optimization process only depends on the dot product of the samples.

   Many real world problems involve nonlinear data sets in which the above methodology will fail. To tackle the non-linearity, 15   using a non-linear function $\Phi(x)$ the feature space is mapped into higher dimensional feature space. The Lagrangian function can be re-written as:

$$L(w, b, \alpha) = \sum_i \alpha_i - \frac{1}{2}\sum_i \sum_j \alpha_i \alpha_j y_i y_j k(x_i, x_j) \tag{7}$$

$$k(x_i, x_j) = \Phi(x_i)^\intercal \Phi(x_j) \tag{8}$$

where $k(x_i, x_j)$ is known as the kernel function. Kernel functions let the feature space be mapped into higher dimensional 20   space without the need of calculating the transformation function (only the kernel is needed). More details on SVM and kernel functions can be found in Bishop (2006)

   To use SVM as a multi-class classifier, some adjustments need to be made to the simple SVM binary model. Methods like a directed acyclic graph, one-against-all, and one-against-others are among the most successful techniques for multi-class classification. Details about these methods can be found in Knerr et al. (1990).

25   **2.3   Decision Trees Algorithms**

Decision trees are nonparametric algorithms that allow complex relations between inputs and outputs, to be modeled. Moreover, they are the foundation of both random forest and boosting methods. A comprehensive introduction to the topic can be found in Quinlan (1986), here, we briefly describe how a decision tree is built.

A decision tree is a set of (binary) decisions represented by an acyclic graph directed outward from a root node to each leaf. Each node has one parent (except the root), and can have two children. A node with no children is called a leaf. Decision trees can be complex depending on the data set. A tree can be simplified by pruning, which means leaves from the upper parts of the trees will be cut. To grow a decision tree, the following steps are taken.

– Defining a set of candidate splits: We should answer a question about the value of a selected input feature to split the data set into two groups.

– Evaluating the splits. Using a score measure, at each node, we can decide what the best question is to be asked and what the best feature is to be used. As the goal of splitting is to find the purest learning subset that is in each leaf, we want the output labels to be the same; called purifying. Shannon Entropy (see below) is used to evaluate the purity of each

subgroup. Thus, a split that reduces the entropy from one node to its descendent is favorable

– Deciding to stop splitting. We set rules to define when the splitting should be stopped, and a node becomes a leaf. This decision can be data-driven. For example, we can stop splitting when all objects in a node have the same label (pure node). The decision can be defined by a user as well. For example, we can limit the maximum depth of the tree (length of the path between root and a leaf).

In a decision tree, by performing a full scan of attribute space the optimal split (at each local node) is selected, and irrelevant attributes are discarded. This method allows us to identify the attributes that are most important in our decision-making process.

The metric used to judge the quality of the tree splitting is Shannon entropy (Shannon, 1948). Shannon Entropy describes the amount of information gained with each event and is calculated as follows:

$$H(x) = -\Sigma p_i \log p_i \tag{9}$$

where $p_i$ represents a set of probabilities that adds up to 1. $H(x) = 0$ means that no new information was gained in the process of splitting, and $H(x) = 1$ means that maximum amount of information was achieved. Ideally, the produced leaves will be pure and have low entropy (meaning all of the objects in the leaf are the same).

## 2.4  Random Forests

The Random forest (RF) method is based on "growing" an ensemble of decision trees that vote for the most popular class.
Typically the bagging (bootstrap aggregating) method is used to generate the ensemble of trees (Breiman, 2002). In bagging, to grow the $k_{th}$ tree, a random vector $\theta_k$ from the training set is selected. The $\theta_{\mathbf{k}}$ vector is independent of the past vectors $(\theta_{\mathbf{1}}, ..., \theta_{\mathbf{k-1}})$ but has the same distribution. Then, by selecting random features, the $k_{th}$ tree is generated. Each tree is a classifier $(h(\theta_k, \mathbf{x}))$ that casts a vote. During the construction of decision trees, in each interior node, the Gini index is used to evaluate the subset of selected features. The Gini index is the measure of impurity of data (Lerman and Yitzhaki, 1984; Liaw
et al., 2002). Thus, it is desired to select a feature that results in a greater decrease in the Gini index (partitioning the data into distinct classes). For a split at node n the index can be calculated as: $1 - \Sigma_{i=1}^{2} P_i^2$ where $P_i$ is the frequency of class $i$ in the node $n$. Finally, the class label is determined via majority voting among all the trees (Liaw et al., 2002).

One major problem in ML is when the algorithm becomes too complicated and perfectly fits the training data points, it looses its generality and performs poorly on the testing set. This problem is known as overfitting. For RF, increasing the number of trees can help with the overfitting problem. Other parameter that can significantly influence RFs is the tree depth, growing more trees in a forest yield a smaller prediction error. Finding the optimal depth of each tree is a critical parameter. While leaves in a short tree may contain heterogeneous data (the leaves are not pure), tall trees can suffer from poor generalization (overfitting problem). Thus, the optimal depth provides a tree with pure leaves and great generalization. Detailed discussion on the RFs can be found in Liaw et al. (2002).

## 2.5 Gradient Boosting Tree Methods

Boosting methods are based on the idea that combining many "weak" approximation models (a learning algorithm that is slightly more accurate than 50%) will eventually boost the predictive performance (Knerr et al., 1990; Schapire, 1990). Thus, many "local rules" are combined to produce highly accurate models.

In the gradient boosting method, simple parametrized models (base models) are sequentially fitted to current residuals (known as pseudo-residuals) at each iteration. The residuals are the gradients of the loss function (they show the difference between the predicted value and the true value) that we are trying to minimize. The Gradient Boosting Trees (GBT) algorithm is a sequence of simple trees generated such that each successive tree is grown based on the prediction residual of the preceding tree with the goal of reducing the new residual. This "additive weighted expansion" of trees will eventually become a strong classifier (Knerr et al., 1990). This method can be successfully used even when the relation between the instances and output values are complex. Compared to the RF model, which is based on building many independent models and combining them (using some averaging techniques), the gradient boosting method is based on building sequential models.

Although, the GBTs show overall high performance, they require large set of training data and the method is quite susceptible to noise. Thus, for smaller training data set the algorithm suffers from overfitting. The size of our data is large, GBT can be potentially a reliable algorithm for classification of lidar profiles.

## 2.6 The t-distributed Stochastic Neighbour Embedding Method

A detailed description of unsupervised learning can be found in (Hastie et al., 2009). Here, we briefly introduce two of the unsupervised algorithms that are used in this paper. The t-SNE method is an unsupervised ML algorithm that is based on Stochastic Neighbor Embedding (SNE). In the SNE, the data points are placed into a low-dimensional space such that the neighborhood identity of each data point is preserved (Hinton and Roweis, 2003). The SNE is based on finding the probability that data point ($i$) has data point ($j$) as its neighbor, which can formally be written as:

$$P_{i,j} = \frac{exp(-d_{i,j}^2)}{\sum_{k \neq i} exp(-d_{i,k}^2)} \tag{10}$$

where $P_{i,j}$ is the probability of $i$ selecting $j$ as its neighbour and $d_{i,j}^2$ is the squared Euclidean distance between two points in the high dimensional space, and can be written as:

$$d_{i,j}^2 = \frac{|| (x_i - x_j) ||^2}{2\sigma_i^2} \tag{11}$$

where $\sigma_i$ is defined so that the entropy of the distribution becomes $\log \kappa$, and $\kappa$ is the "perplexity", which is set by the user and determines how many neighbors will be around a selected point.

The SNE tries to model each data point, $x_i$, at the higher dimension, by a point $y_i$ at a lower dimension such that the similarities in $P_{i,j}$ are conserved. In this low dimensional map, we assume that the points follow a Gaussian distribution. Thus, the SNE tries to make the best match between the original distribution $(p_{i,j})$ and the induced probability distribution $(q_{i,j})$. This match is determined by minimizing the error between the two distributions, and the best match is developed. The induced probability is defined as:

$$q_{i,j} = \frac{exp(- || (y_i - y_j) ||^2)}{\sum_{k \neq i} exp(- || (y_i - y_k) ||^2)} \tag{12}$$

The SNE algorithm aims to find a low-dimensional data representation such that the mismatch between $p_{i,j}$ and $q_{i,j}$ become minimized; thus in the SNE the Kullback-Leibler divergences is defined as the cost function. Using the gradient descent method the cost function is minimized. The cost function is written as:

$$cost = \Sigma_i KL(P_i || Q_i) = \Sigma_i \Sigma_j p_{j|i} \log \frac{p_{j|i}}{q_{j|i}} \tag{13}$$

where $P_i$ is the conditional probability distribution of all data points given data points $x_i$, and $Q_i$ is the conditional probability for all the data points given data points $y_i$.

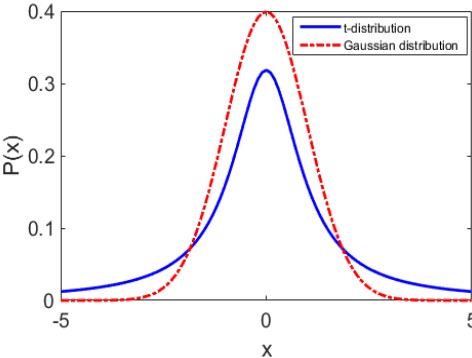

**Figure 2.** Red curve: the Gaussian distribution for data points, extending from -5 $\sigma$ to 5$\sigma$. The mean of the distribution is at 0. Blue curve: The Student's t-distribution over the same range. The distribution is heavy-tailed, compared to the Gaussian distribution.

The t-SNE uses a similar approach but assumes a lower dimensional space, which instead of being a Gaussian distribution follows Student's t-distribution with a single degree of freedom. Thus, since a heavy-tailed distribution is used to measure

similarities between the points in the lower dimension, the data points that are less similar will be located further from each other. To demonstrate the difference between the Student's t-distribution and the Gaussian distribution, we plot the two distributions in Fig. 2. Here, the $x$ values are within 5 $\sigma$ and -5 $\sigma$. The Gaussian distribution with the mean at 0 and the Student's t-distribution with the degree of freedom of 1 are generated. As is shown in the figure, the t-distribution peaks at a lower value

and has a more pronounced tail. The above approach gives t-SNE an excellent capability for visualizing data, and thus, we use this method to allow scan classification via unsupervised learning. More details on SNE and t-SNE can be found in Hinton and Roweis (2003) and Maaten and Hinton (2008).

## 2.7 Density-based spatial clustering of applications with noise (DBSCAN)

DBSCAN is an unsupervised learning method that relies on density based clustering and is capable of discovering any arbitrary

shape from a collection of points. There are 2 input parameters to be set by the user: minPts, which indicates the minimum number of needed oints to make a cluster, and $\epsilon$ such that the $\epsilon$ neighborhood of point $p$ denoting as $N_\epsilon(p)$ is defined as:

$$N_\epsilon(p) = \{q \in D | dis(p, q) \leq \epsilon\} \tag{14}$$

where $p$ and $q$ are two points in data set $(D)$ and $dist(p, q)$ represents any distance function. Defining the 2 input parameters, we can make clusters. In the clustering process data points are classified into 3 groups: core points, (density) reachable points

and outliers define as follows.

- – Core point: point $A$ is a core point if within the distance of $\epsilon$ at least minPts points (including $A$) exist.

- – Reachable point: point $B$ is reachable from point $A$ if there is a path $(P_1, P_2,..., P_n)$ from $A$ to $B$ $(P_1 = A)$. All points in the path, with the possible exception of point $B$, are core points.

- – Outlier point: point $C$ is an outlier if it is not reachable from any point.

In this method, an arbitrary point (that has not been visited before) is selected, and using the above steps the neighbor points are retrieved. If the created cluster has sufficient number of point (larger than minPts) a cluster is started. One advantage of DBSCAN is that the method can automatically estimate the numbers of clusters.

## 2.8 Hyper-parameter tuning

Machine learning methods are generally parametrized by a set of hyper-parameters, $\lambda$. An optimal set $\lambda_{best}$ will result in an

optimal algorithm which minimizes the loss function. This set can be formally written as:

$$\lambda_{best} = \arg\min\{L(X_{test}; A(X_{train}, \lambda))\} \tag{15}$$

where $A$ is the algorithm and $X_{test}$ and $X_{train}$ are test and training data. Searching to find the best set of hyper-parameters is mostly done by grid search method in which a set of values on a predefined grid is proposed. Implementing each of the proposed hyperparameters the algorithm will be trained, and the prediction results will be compared. Most algorithms have

| Test set | | | | Training set | | | |
|---|---|---|---|---|---|---|---|
| Channel | SVM | RF | GBT | Channel | SVM | RF | GBT |
| HR | 83% | 97% | 98% | HR | 90% | 98% | 99% |
| LR | 88% | 97% | 97% | LR | 90% | 98% | 98% |
| Nitrogen | 88% | 95% | 96% | Nitrogen | 88% | 94% | 95% |

**Table 1.** Accuracy scores for the training and the test set for SVM, RF, and GBT models. Results are shown for HR, LR, and nitrogen channels.

only few hyper-parameters. Depending on the learning algorithm, the size of training and test data sets the grid search can be a time consuming approach. Thus automatic hyper-parameter optimization has gain interest, details on the topic can be found in Feurer and Hutter (2019).

## 3 Result for supervised and unsupervised learning using the PCL system

### 3.1 Supervised ML Results

To apply supervised learning to the PCL system, we randomly chose 4500 profiles from the LR, HR and the nitrogen vibrational Raman channels. These measurement were taken on different nights in different years and represent different atmospheric conditions. For the LR and HR digital Rayleigh channels, the profiles were labeled as "bad profiles" and "good profiles". For the nitrogen channel we added one more label that represents profiles with traces of clouds or aerosol layers, called "cloudy" profiles. Here, by "cloud" we mean a substantial increase in scattering relative to a clean atmosphere, which could be caused by clouds or aerosol layers. The HR and LR channels seldom are affected by clouds or aerosols as the chopper is not fully open until about 20 km. Furthermore, labeling the water vapour channel was not attempted for this study, due to its high natural variability in addition to instrumental variability.

We used 70% of our data for the training phase and we kept 30% of data for the test phase (meaning that during the training phase 30% of data stayed isolated and the algorithm was built without considering features of the test data). In order to overcome the overfitting issue we used the $k$-fold cross-validation technique in that the data set is divided into $k$ equal subsets. In this work we used 5-fold cross-validation. The accuracy score is the ratio of correct predictions to the total number of predictions. We used accuracy as a metric of evaluating the performance of the algorithms. We used the Python scikit-learn package to train our ML models. The prediction scores resulting from the cross validation method as well as from fitting the models on the test data set is shown in Table 1.

We also used the confusion matrix for further evaluations where the good profiles are considered as "positive" and the bad profiles are considered as "negative". A confusion matrix can provide us with the number of:

– True positives (TP): number of profiles that are correctly labeled as positive (clean profiles)

– False positives (FP): number of profiles that are incorrectly labeled as positive

| Nitrogen channel | | |
| --- | --- | --- |
| Scan Type | Precision | Recall |
| Cloud | 0.94 | 0.91 |
| Clear | 0.96 | 0.98 |
| Bad | 1.00 | 1.00 |

| LR channel | | |
| --- | --- | --- |
| Scan Type | Precision | Recall |
| Clear | 0.99 | 0.99 |
| Bad | 0.96 | 0 .95 |

| HR channel | | |
| --- | --- | --- |
| Scan Type | Precision | Recall |
| Clear | 0.98 | 1.00 |
| Bad | 0.98 | 0.94 |

**Table 2.** Precision and recall values for the nitrogen, LR, and HR channels. The precision and recall values are calculated using the GBT model.

- True negatives (TN): number of profiles that are correctly labeled as negative (bad profiles)

- False negatives (FN): number of profiles that are incorrectly labeled as negative.

A perfect algorithm will result in a confusion matrix in that FP and FN are zeros. Moreover, the precision and recall can be employed to give us an insight on how our algorithm can distinguish between good and bad profiles. The precision and recall are defined as follows:

$$Precision = \frac{TruePositive}{TruePositive + FalsePositive} \qquad Recall = \frac{TruePositive}{TruePositive + FalseNegative} \qquad (16)$$

The precision and recall for the nitrogen channel for each category (clear, cloud, and bad) are shown in Table 2 as well. The GBT and RF algorithms, both have high accuracy results on HR and LR channels. The accuracy of the model on the training set on the LR channel for both RF and GBT are 99% and on the test set are 98%. The precision and recall values for the clears profiles are close to unity and for the bad profiles they are 0.95 and 0.96 respectively. The HR channel also has a high accuracy of 99% in the training set for both RF and GBT, and the accuracy score in the testing set is 98%. The precision and recall values in Table 2 are also similar to the LR channel.

For the nitrogen channel the GBT algorithm has the highest accuracy of 95% while the RF algorithm with accuracy of 94%. The confusion matrix of the test result for the GBT algorithm (the one with the highest accuracy) is shown in Fig. 3 (left panel). The algorithm can perform almost perfectly on distinguishing bad profiles (only one bad scan was wrongly labeled as cloudy). The cloud and clear profiles for most profiles are labeled correctly; however for few profiles the model mislabeled clouds for clear profiles.

### 3.2 Unsupervised ML Results

The t-SNE algorithm clusters the measurements by means of pairwise similarity. The clustering can differ from night to night due to atmospheric and systematic variability. On nights where most profiles are similar, fewer clusters are seen, and on other nights when the atmospheric or the instrument conditions are more variable, more clusters are generated. The t-SNE makes clusters but does not estimate how many clusters are built, and the user must then estimate the number of clusters. To automate

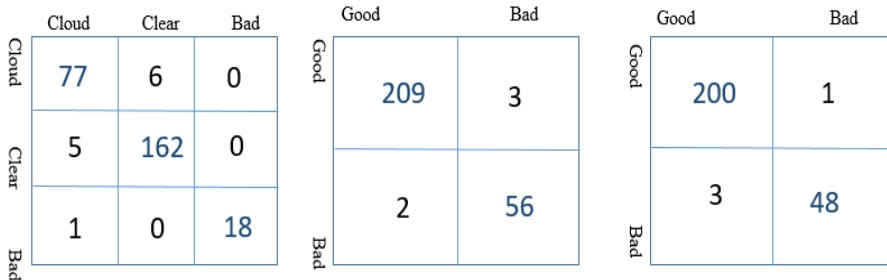

**Figure 3.** The confusion matrices for nitrogen channel (left), LR channel (middle), and HR channel (right). In a perfect model, the off-diagonal elements of the confusion matrix are zeros.

this procedure, after applying the t-SNE to lidar measurements we use the DBSCAN algorithm to estimate the number of clusters. The second step of applying DBSCAN is used to estimate the number of generated clusters by t-SNE.

To demonstrate how clustering works, we show measurements from the PCL LR channel and the nitrogen channels. Here, using the t-SNE on May 15, 2012 that contains both bad and good profiles. We also show the clustering result for the nitrogen channel on May 26 2012. We chose this night because at the beginning of the measurements the sky was clear but the sky became cloudy.

On the night of May 15 2012, the t-SNE algorithm generates three distinct clusters for the LR channel (Fig. 4). These cluster correspond to different types of lidar return profiles. Figure 5 (left panel) shows all the signals for each of the clusters. The maximum number of photon counts and the value and the height of the background counts are the identifiers between different clusters. Thus, cluster 3 with low background counts and high maximum counts represents a group of profiles which are labeled as good profiles in our supervised algorithms. Cluster 1 represents the profiles with lower than normal laser powers, and clusters 2 shows profiles with extremely low laser powers. To better understand the difference between these clusters, Fig. 5 (right panel) shows the average signal. Furthermore, the outliers of cluster 3 (shown in black) identify the profiles with extremely high background counts. This result is consistent with our supervised method, in which we had good profiles (here is cluster 3), and bad profiles which are profiles with lower laser power (here are cluster 1 and 2).

Using the t-SNE, we also have clustered profiles for the nitrogen channel with the measurements taken on May 26, 2012. This night was selected because the sky conditions changed from clear to cloudy. The measurements from this night allows us to test our algorithm and determine how well it can distinguish cloudy profiles from the non-cloudy profiles. The result of clustering is shown in Fig.6 (left panel) in that two well-distinguished clusters are generated, where one cluster represents, the cloudy and the other represents the non-cloudy profiles. The averaged signal for each cluster is plotted in Fig.6 (middle panel). Moreover, the particle extinction profile at altitudes between 3 km to 10 km is plotted in the same figure (Doucet, 2009). The first 130 profiles are clean and the last 70 profiles are severely affected by thick clouds; thus the extinction profile is consistent with our t-SNE classification result.

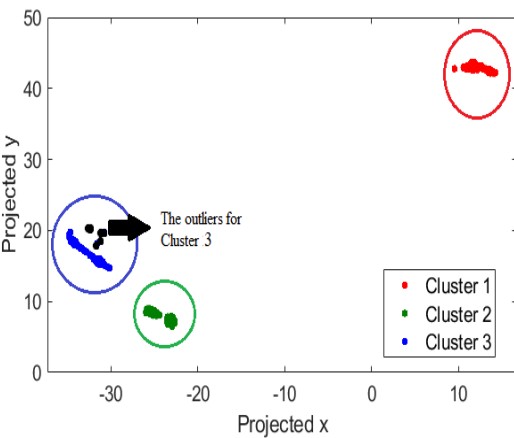

**Figure 4.** Clustering of lidar return signal type using the t-SNE algorithm for 339 profiles from the low-gain Rayleigh measurement channel on the night of May 15 2012. The profiles are automatically clustered into three different groups selected by the algorithm. Cluster number 3 has some outliers.

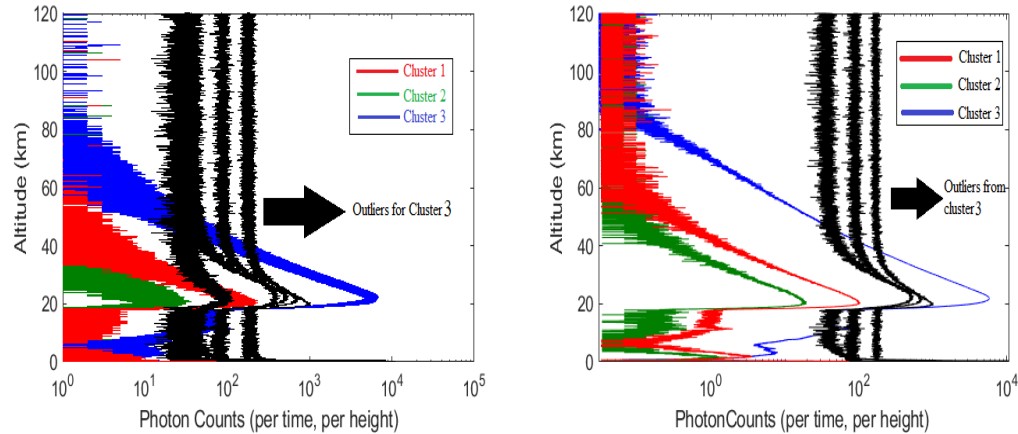

**Figure 5.** Left Panel: All 339 profiles collected by the PCL system LR channel on the night of May 15, 2012. The sharp cutoff for all profiles below 20 km is due to the system's mechanical chopper. The green signals have extremely low power. The red line represents all signals with low return signal and the blue line indicates the signals that are considered good profiles. The black lines are signals with extremely high backgrounds. Right Panel: Each line represents an average of the signals within a cluster. The red line is the average signal for profiles with lower laser power (cluster 1). The green line is the average signal for profiles with really low laser power (cluster 2). The blue line is the average signal for profiles with strong laser power (cluster 3). The black line indicates the outliers that have extremely high background counts and are outliers belonging to cluster 3 (blue curve). The background counts in the green line start at about 50 km, where as for the red line the background starts at almost 70 km while for the blue line profiles the background starts at 90 km.

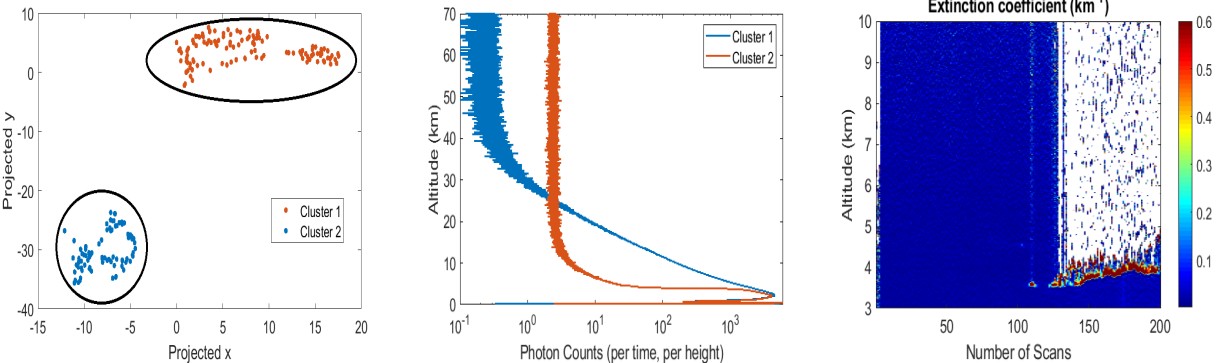

**Figure 6.** Left panel: profiles for the Nitrogen channel on the night of May 15, 2012 were clustered into two different groups using the t-SNE algorithm. Middle panel: The red line (cluster number 2) is the average of all signals within this cluster and indicates the profiles in that clouds are detectable. The blue line (cluster number 2) is the average of all signals within this cluster and indicates the clear profiles (non-cloudy condition). Right panel: the particle extinction profile for the night shows the last 70 profiles are affected by thick clouds at about 4.5 km altitude.

The t-SNE method can be used as a visualization tool; however to evaluate each cluster the user either needs to examine profiles within each cluster or use one of the mentioned classification methods. For example Fig. 4 shows this night of measurement had some major differences among the collected profiles (if all the profiles were similar only one cluster would be generated). But, to evaluate the cluster the profiles within each cluster must be examined by a human or a supervised ML should be used to label each cluster.

## 3.3  PCL fire detection using the t-SNE algorithm

The t-SNE can be used for anomaly detection. As fire's smoke in the stratosphere is relatively a rare event, we can test the algorithms to identify these events. Here, we used the t-SNE to explore traces of aerosol in stratosphere within one month of measurements. We expect that the t-SNE would generate a single cluster for a month with no trace of stratospheric aerosols that means no "anomalies" has been detected. The algorithm should generate more than one cluster in the case of detecting stratospheric aerosols. We use DBSCAN algorithm to automatically estimate the number of generated profiles. In DBSCAN, most of the bad profiles will be tagged as noise (meaning that they do not belong to any cluster). Here we are showing two examples, in one of which the stratospheric smoke exists and our algorithm generates more than one cluster. In the other example stratospheric smoke is not present in the profiles, and the algorithm only generates one cluster. The nightly measurements of June 2002, are used as an example of a month in which the t-SNE can detect anomalies (thus more than one cluster is generated), as the lidar measurement were affected by the wildfire in Saskatchewan, and nights of measurements in July 2007 are used as an example of nights with no high loads of aerosol in stratosphere (only one cluster is generated).

The wildfires in Saskatchewan during late June and early July 2002 produced massive smoke that was transported southward. As the smoke from the fire can reach to higher altitudes (reaching to lower stratosphere), we are interested to see if we can

automatically detect stratospheric aerosol layers during wildfire events. The PCL was operational on the nights of June 8, 9, 10, 19, 21, 29 and 30 of 2002. During these nights, 1961 lidar profiles were collected in the nitrogen channel. We used the t-SNE algorithm to examine if the algorithm can detect and cluster the profiles with the trace of wildfire in higher altitudes; using profiles in the altitude range of 8 to 25 km. To automatically estimate the number of produced clusters we used the

DBSCAN algorithm. We set the minPts condition to 30, and the $\epsilon$ value to 3. The DBSCAN algorithm estimated 4 clusters and few profiles remained as the noise, which do not belong to any other clusters (shown in cyan, in Fig. 7). To investigate if profiles with layers are clustered together the particle extinction profile for each generated cluster is plotted (Fig. 8). Most of the profiles in cluster 1 are clean and no sign of particles can be seen in these profiles (top-left panel in Fig. 8), cluster 2 contains all profiles with mostly small traces of aerosol between 10 km to 14 km (top-right panel in Fig. 8). the presence of

high loads of aerosol can clearly detected in the particle extinction profiles for both cluster 3 and 4; the difference between the two cluster is in the height in which the presence of aerosol layer is more distinguished (bottom-right and bottom-left panel in Fig. 8). profiles in the last two clusters belong to the last two nights of measurements on June 2002 which are coincidental with the smoke being transported to London from the wildfire in Saskatchewan.

    We also examined the total of 2637 profiles in the altitude range of 8 km to 25 km obtained from 10 nights of measurements

in July 2007. As we expected, no anomalies were detected (Fig. 9). The particle extinction profile of July 2007 also indicates that at the altitude range of 8 km to 25 km no aerosol load exists (Fig. 9).

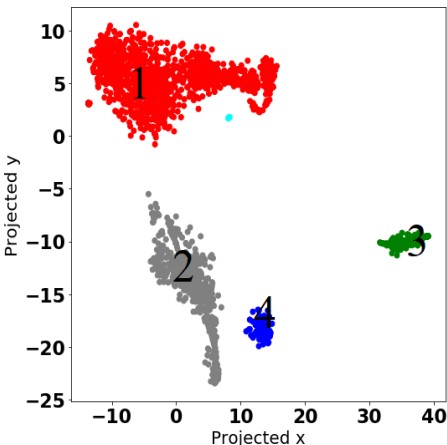

**Figure 7.** profiles for the Nitrogen channel for the nights of June 2002 were clustered into four different groups using the t-SNE algorithm. The small cluster in cyan indicates the group of profiles that do not belong to any of the other clusters in the DBSCAN algorithm.

    Thus, using the t-SNE method we can detect anomalies in the UTLS. In the UTLS region, for the clear atmosphere we expect to see a single cluster; and when aerosol loads exist at least two clusters will be generated. We are implementing the t-SNE on one month of measurements, and when the algorithm generates more than one cluster we examine profiles within that cluster.

However, at the moment, because we only use the Raman channel it is not possible for us to distinguish between smoke traces

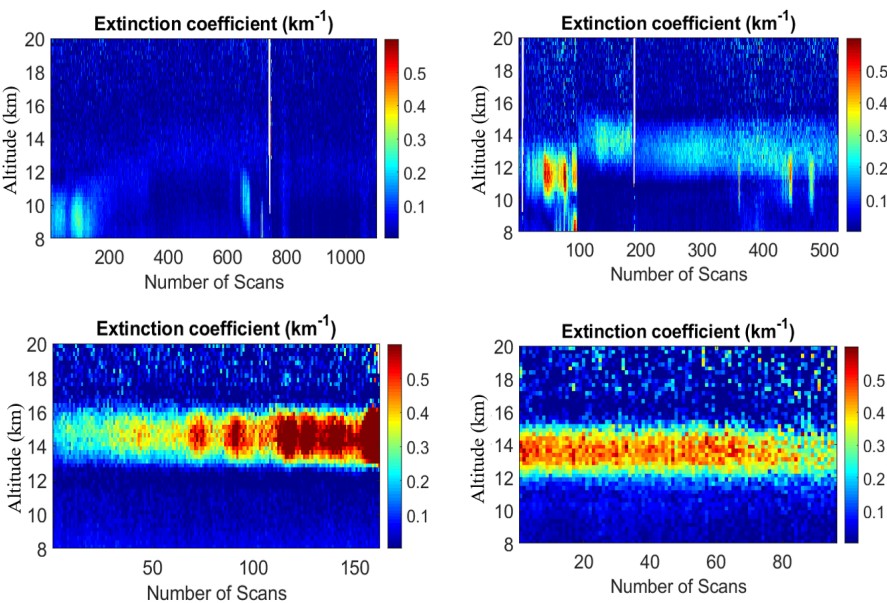

**Figure 8.** The t-SNE generates four clusters for profiles of nitrogen channel in July 2007. Top-left panel: Most of the profiles are clean and no sign of particles can be seen. Top-right panel: profiles with mostly small traces of aerosol between 10 km to 14 km. Bottom panels: the presence of high loads of aerosol can clearly be detected.

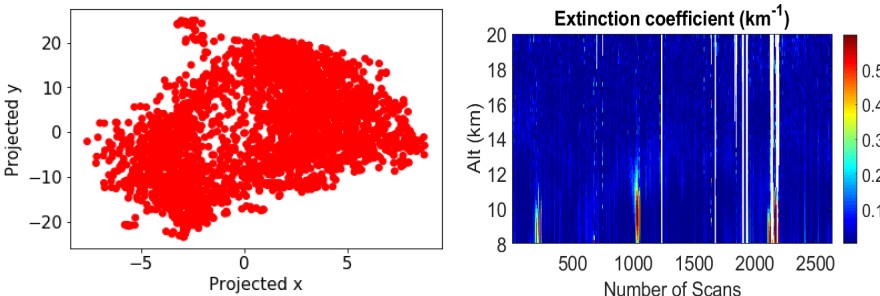

**Figure 9.** Right panel: The t-SNE generates a single cluster for all of 2637 profiles of nitrogen channel for July 2007. Left panel: The particle extinction profile of July 2007 indicates no significant trace of stratospheric aerosols.

and cirrus clouds (unless the trace is detected in altitudes above 14 km; similar to fig. 8 where we are more confident to claim that the detected aerosol layers are traces of smoke, as shown in Fromm et al. (2010)).

## 4    Summary and Conclusion

We introduced a machine learning method to classify raw lidar (level-0) measurements. We used different ML methods on elastic and inelastic measurements from the PCL lidar systems. The ML methods we used and our results are summarized as follows.

1. We tested different supervised ML algorithms, among which the RF and the GBT performed better, with a success rate above 90% for the PCL system.

2. The t-SNE unsupervised algorithm can successfully cluster profiles on nights with both consistent and varying lidar profiles due to both atmospheric conditions and system alignment/performance. For example, if during the measurements the laser power dropped or clouds became present, the t-SNE showed different clusters representing these conditions.

3. Unlike the traditional method of defining a fixed threshold for the background counts, in supervised ML approach the machine can distinguish high background counts by looking at the labels of the training set. In the unsupervised ML approach, by looking at the similarities between the two profiles and defining a distance scale; good profiles will be grouped together. High background counts can be grouped in a smaller group. Most of the time the number of bad profiles are small, thus they will be labeled as noise.

We successfully implemented supervised and unsupervised ML algorithms to classify lidar measurement profiles. The ML is a robust method with high accuracy that enables us to precisely classify thousands of lidar profiles within a short period of time. Thus, with accuracy of higher than 95% this method has a significant advantage over previous methods of classifying. For example, in the supervised ML, we train the machine by showing (labeling) different profiles in different conditions. When the machine has seen enough examples of each class (that is a small fraction of the entire data base), it can classify the un-labeled profiles with no need to pre-define any condition for the system. Furthermore, in the unsupervised learning method, no labeling is needed, and the whole classification is free from subjective biases of the individual marking the profiles (which for large atmospheric data sets ranging over decades is important). Using ML avoids the problem of different observers classifying profiles differently. We also showed that the unsupervised schema has the potential to be used as an anomalies detector; which can alert us when there is a trace of aerosol in the UTLS region. We are planning to expand our unsupervised learning method to both Rayleigh and Nitrogen channels to be able to correctly identify and distinguish cirrus clouds from smoke traces in the UTLS. Our results indicate that ML is a powerful technique that can be used in lidar classifications. We encourage our colleagues in the lidar community to use both supervised and unsupervised ML algorithms for their lidar profiles. For the supervised learning the GBT performs exceptionally well, and the unsupervised learning has the potential of sorting anomalies.

*Acknowledgements.* We would like to thank Sepideh Farsinejad for many interesting discussions about clustering methods and statistics. We would like to thank Shayamila Mahagammulla Gamage for her inputs on labeling methods; and Robin Wing for our chats about other

statistical methods used for lidar classification. We also recognize Dakota Cecil's insights and help on the labeling process of the raw lidar profiles.

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
