# Peer review of "Classification of Lidar Measurements Using Supervised and Unsupervised Machine Learning Methods"

_Atmospheric Measurement Techniques, 2019_

## Short Comment (SC1) · 27 May 2020

This paper presents a classification of raw lidar signals using machine learning techniques.

I have a few questions regarding the methods the authors chose and results:

1) I understand that for the supervised training around a few thousand scans were selected, and photon counts at altitudes were used as features. How many features did you have? (what is the dimension of the training set)

2) I realize that t-SNE is a strong unsupervised method, but much slower than some

other techniques such as KMeans clustering. Is that any reason that you have not implemented KMeans method?

---

## Author Comment (AC1) · 15 Jun 2020

1) I understand that for the supervised training around a few thousand scans were selected, and photon counts at altitudes were used as features. How many features did you have? (what is the dimension of the training set)

For the Rayleigh channels we have 2300 features as the lidar samples a broad range of altitudes, 25\km to 110 \km. For the nitrogen channel, the number of features decreases to 300 as the Raman scattering is weaker and above 25 \km altitude the signal to noise ratio drops. The number of features is sufficient in both cases, as they provide acceptable results, meaning using these features we can successfully train and

predict classifications for lidar scans.

2) I realize that t-SNE is a strong unsupervised method, but much slower than some other techniques such as KMeans clustering. Is that any reason that you have not implemented KMeans method?

The concept of KMeans clustering is based on finding spherical clusters which have a defined centroid for each cluster. Moreover, the number of clusters is set as an input parameter, and a wrong number of predefined clusters can result in unphysical results. KMean clustering is a reasonable way to proceed when there are principal physical arguments to hypothesize a specific number of categories/clusters. However, as we are performing exploratory data analysis (we might or might not encounter scans with traces of fire smoke), it is preferable to use a technique that "lets the data speak for itself", in which case t-SNE is a better choice of methods.

---

## Referee Comment (RC1) · Benoît Crouzy (Referee) · 8 Sep 2020

The authors apply various machine learning techniques (ML) to classify Lidar measurements. They show the potential of different approaches: supervised technique in the presence of known categories and unsupervised techniques in order to detect anomalies in the signal resulting from unusual events (e.g. fires). The paper begins with an extensive introduction on machine learning which could be useful to the Lidar community, and the chosen applications show a good panel of possibilities to apply ML on lidar measurements.

Below I list my comments in the order as they appear in the manuscript.

[Figure]

1) Page 1 line 10: selection of the training and test set. This is a potential major issue. The authors selected randomly scans over the years. In order to have good generalization of the algorithms it however is important to have not too close training and test data. The total number of raw scans out of which events were randomly drawn suggests that this is achieved but I would like the authors to comment on this question, even if only as a caveat to the community. Best practice could be to select a period isolated from the training set to select the test set (e.g. different year).

2) Page 2 lines 3-8: please check for some repetitions (laser fluctuations)

3) Page 2 line 17-18: I suggest to distinguish between supervised and non-supervised from the onset.

4) Page 2 line 26: I would put this line earlier to distinguish between supervised and non-supervised techniques. In addition I would hint for non-specialists that unsupervised techniques are no silver bullet and can be expected to be less powerful due to the absence of training data.

5) Page 2 line 27: "clustering ML" and line 30 "These ML methods" please precise which methods.

6) Page 3 line 23: I find the sigma confusing (summation sign), especially when keeping signed differences.

7) Page 3 line 25: what about "matrix size (m,n)" or m x n ?

8) Page 4 line 13-14: please describe better the Kernel trick that make SVM so powerful. In the current form I do not find the description self-contained (non-uniform level of details). I would describe the parameters to be tuned (e.g. Cost, epsilon insensitive tube, ...)

9) Page 4 line 25: please define all variables

10) Page 5 line 3: maybe I missed it, but if you used LIBSVM or a derived tool please

mention it, as this information could help other users.

11) Page 5 line 25: I found this sentence somewhat disconnected.

12) Equation 3: define the class index.

13)Page 5 line 29: H=0 usually means low entropy which could be seen as a pure prediction. Please clarify.

14) General comment: I would summarize for all methods the hyperparameters to be tuned. This is currently well done only for some of the methods.

15) Page 7 line 1: "a detailed description" this sentence and the introduction to various methods give the impression of ML as a closed list of techniques. As the Lidar community is not very familiar to ML, I would mention that a vast number of other techniques exist. I would also mention ANNs and explain why those were not used in the present paper.

16) I would remove Figure 2 (too much detail in comparison with the rest of the chapter), but this is a matter of personal taste.

17) Page 8 line 15: from and not form

18) Section 3.1: see general comment above, how was it ensured that enough separation between training situations and test situations is achieved (scans taken the same day/hour might not always achieve this). This point needs to be discussed carefully.

19) Section 3.2: In my opinion the discussion on TP/FP/TN/FN does not belong to the results but to the methods.

20) Page 11 line 6: to estimate

21) Page 14 line 11: anomalies and not anomolies

22) General comment: why not making use of thresholds in order to achieve finer objectives, eg. "never tagging good scans as bad"? For example, with SVMs one can

use the distance to the hyperplane to select events with a good likelihood of correct classification.

In all, I recommend the paper for publication after the questions raised are satisfactorily addressed by the authors.

---

## Referee Comment (RC2) · Anonymous Referee #2 · 9 Sep 2020

[journal abbreviation, manuscript]article

etoolbox mathptmx [11pt]moresize blindtext, xfrac

**General comment**

The study "Classification of Lidar Measurements Using Supervised and Unsupervised Machine Learning Methods" by Ghazal Farhani, Robert J. Sica, and Mark Joseph Daley deals with the important topic of automatic selection of "good" and "bad" data. The application is on Raman and Rayleigh lidar signals in cloudy, clear and aerosol-loaded conditions. The presented methodology is based on machine learning (ML) technique (both supervised and unsupervised) and allows to efficiently creating clusters of data

to facilitate the processing of the lidar data and to set up an alert system. The study presents first an overview of the most representative techniques using ML concept and then provides real examples of supervised and unsupervised techniques applied to the PCL signal. The topic is without any doubt of high interest, as it provides a solid solution to improve the representation and interpretation of lidar data.

The ML algorithm overview is comprehensive and well structured, as well as the part dedicated to the real examples from the PCL. However, starting from Section 3 the description and discussion of the tests results become somewhat confused, with an increasing number of typos and grammar mistakes that make the comprehension more difficult. Especially the discussion around the results shown in Figures 7-9 struggles to get to the point and loses in effectiveness to highlight the strengths of the t-SNE and DBSCAN methods. I recommend the authors to pay special attention to improve this part.

Despite these imperfections, I support the publication of this study as scientific article in AMT after revision.

**Technical comments**

*Throughout the text*

1. Why the authors use the term *scan* to indicate a lidar profile? I am used to refer to a scan strictly as for a scanning device, e.g., a scanning wind lidar will give a profile of wind strength and direction. The recording in time of the photon-counting signal when transformed into altitude is better called a profile.

2. When the authors use the relative pronoun "which", this shall come after a comma otherwise "that" has to be used instead.

3. Plenty of punctuation, articles and auxiliary verbs are missing through the text. I have tried to highlight a part of them, but the authors should perform a thorough re-reading of the manuscript and correct these typos and errors.

Sect.1, Pg 2, ln 20: the PCL acronym has already been defined.

Sect.2, Pg 3, ln. 8: you can remove "which are as follows".

Sect.2, Pg 3, ln. 23: "minimizes"

Figure 1, caption: I'd say the line color in the left panel is Cyan rather than blue.

Sect. 2.2, Pg 4, ln 5: "we have tested" (remove "been")

Sect. 2.4, pg6, ln 3: As it is mentioned here for the first time, it could be useful to add the extended name of bagging in brackets "bagging (bootstrap aggregating)" .

Sect. 2.5, pg 6, ln 28-29: the definition of overfitting should be provided in Sect. 2.4 when it is first introduced.

Sect. 3.1, pg 9, ln 4: "….different years AND represent different…"

Sect. 3.1, pg 9, ln 21: change "True negatives (FN)" to "False negative (FN)"

Sect. 3.1, pg 9, last paragraph: first you define precision and recall and then you present results of accuracy in Table 1. I would show Table 2 before Table 1, right below equations 9.

Sect. 3.1, pg 10, ln 5: "clear scans"

Sect. 3.3, pg 13, ln 9: replace "smoke does not present" with "smoke is not present".

Sec. 3.3., pg 13, ln 12: replace "during late June and early June 2002" with "during early June and late June 2002".

Sec. 3.3., pg 13, ln 13: "As, the smoke" without a comma

Sec. 3.3., pg 13, ln 16: "1961 lidar scans"

Sect. 3.3, pg 14, ln 2: replace "To investigate if scans with layers are clustered together the particle extinction …" with "To investigate if scans with layers are clustered together, the particle, extinction…".

Sect. 3.3, pg 14, ln 10-11: please consider rephrasing and correcting the English.

Sect. 3.3, pg 14, ln 11: "no anomalies WERE detected"

Sect. 4, pg 16: "We are planning to expand our unsupervised learning method to both Rayleigh and Nitrogen channels to be able to correctly identify and distinguish cirrus clouds from smoke traces in the UTLS.". Does the PCL have a depolarization channel, or is in the forthcoming plans to implement one to discriminate between different

particles based on their asphericity?

---

## Author Response (AR1)

**Response to Reviewer 1: Benoît Crouzy (Referee):**

**Thanks a lot for your comments here are our responses.**

1) Page 1 line 10: selection of the training and test set. This is a potential major issue. The authors selected randomly scans over the years. In order to have good generalization of the algorithms it however is important to have not too close training and test data. The total number of raw scans out of which events were randomly drawn suggests that this is achieved but I would like the authors to comment on this question, even if only as a caveat to the community. Best practice could be to select a period isolated from the training set to select the test set (e.g. different year).

We agree that setting a period totally isolated for the test is the best practice and can test the generality of a model, but we wanted our training set to include most possible scenarios (different scans). In our training phase we had 3150 scans, and in the testing phase we had 850 scans. As the number of test sets is large (compared to the total number of selected scans), we have confidence in achieving a model capable of generalization. We also recommend that the model be retrained every year (adding new data to the existing pool of training data). Retraining with more (and newer) data will help with the overfitting problem.

**2) Page 2 lines 3-8: please check for some repetitions (laser fluctuations)**

**We shortened the paragraph.**

3) Page 2 line 17-18: I suggest to distinguish between supervised and non-supervised from the onset.

**We changed the paragraph as follows:**

Here, we propose a machine learning (ML) approach for level-0 data classification. The classification of lidar profiles is based on supervised ML techniques which will be discussed in detail in Section 2. Using an unsupervised ML approach, we also have examined the capability of ML to detect anomalies ...

4) I would put this line earlier to distinguish between supervised and non-supervised techniques. In addition I would hint for non-specialists that unsuper- vised techniques are no silver bullet and can be expected to be less powerful due to the absence of training data.

**Done!**

5) Page 2 line 27: "clustering ML" and line 30 "These ML methods" please precise which methods.

We changed the sentence to: Both Zeng et al. (2018) and Nicolae et al. (2018) concluded that their proposed ML algorithms can classify large sets of data and can successfully distinguish between different types of aerosols.

6) Page 3 line 23: I find the sigma confusing (summation sign), especially when keeping signed differences.

We explained the equation with more details:

Formally, we are trying to learn a prediction function f(x): x \rightarrow y\$ which minimize the expectation of some loss function  $L(y,f) = \sum_{i=1}^{i} N(y^{true}_{i} - y^{predicted}_{i})$ , where  $y^{true}_i$  is the actual value (label) of the classification for each data point, and  $y^{predicted}_{i}$  is the prediction generated from the prediction function and N is the length of data-set  $citep{bishop2006pattern}$

7) Page 3 line 25: what about "matrix size (m,n)" or m x n?

We changed to a matrix with size (m,n)

8) Page 4 line 13-14: please describe better the Kernel trick that make SVM so powerful. In the current form I do not find the description self-contained (non-uniform level of details). I would describe the parameters to be tuned (e.g. Cost, epsilon insensitive tube, ...)

We added the following to the paper which explains the SVM in more detail (there is also a colored marked up pdf where you can see the latex below rendered):

\begin{align}

*minimize* &: \frac{1}{2}||\vec{w}||^2 \\

subject &: y\_{i}(\vec{w}.\vec{x\_i} +b) \geqslant 1.

\end{align}

*In the above equation the constraint is a linear model. To solve this constrained optimization problem, the Lagrange function can be built:*

\begin{equation}

 $L(\operatorname{vec}\{w\}, b, \operatorname{alpha}) = \operatorname{frac}\{1\}\{2\} \operatorname{vec}\{w\}^2\} - \operatorname{vec}[w]^2 - \operatorname{vec}[w]^2$

**\end{equation}**

where  $\lambda_i = Lagrangian multipliers. Setting the derivatives of L(\vec{w}, b, \alpha) with respect to <math>\lambda_i = Lagrangian multipliers. Setting the derivatives of L(\vec{w}, b, \alpha) with respect to <math>\lambda_i = Lagrangian multipliers. Setting the derivatives of L(\vec{w}, b, \alpha) alpha) alpha).$

\begin{align}

\vec{w} &= \sum\_i \alpha\_i y\_i \vec{x}\_i \\

& \sum\_i \alpha\_i y\_i = 0.

\end{align}

Thus we can rewrite the Lagrangian as:

\begin{equation}

L(\vec{w}, b, \alpha) = \sum\_i \alpha\_i - \frac{1}{2} \sum\_i \sum\_j \alpha\_i \alpha\_j y\_i y\_j \vec{x}\_i . \vec{x}\_j

\end{equation}

It is clear that the optimization process only depends on the dot product of the samples.

Many real world problems involve nonlinear data sets in which the above methodology will fail. To tackle the non-linearity, using a non-linear function \$\Phi (x)\$ the feature space is mapped into higher dimensional feature space. The Lagrangian function can be re-written as:

\begin{align}

 $L(\operatorname{vec}\{w\}, b, \operatorname{alpha}) &= \operatorname{sum}_i \operatorname{alpha}_i - \operatorname{frac}\{1\}\{2\} \operatorname{sum}_i \operatorname{sum}_j \operatorname{alpha}_i \operatorname{alpha}_j y_i y_j \\ k(\operatorname{vec}\{x\}_i, \operatorname{vec}\{x\}_j) \\ (\operatorname{vec}\{x\}_i, \operatorname{vec}\{x\}_j) \\ (\operatorname{vec}\{x\}_j) \\ (\operatorname{vec}\{x\}_j) \\ (\operatorname{vec}\{x\}$

k(\vec{x}\_i, \vec{x}\_j) &= \Phi(\vec{x}\_i) . \Phi(\vec{x}\_j)

\end{align}

where \$k(\vec{x}\_i, \vec{x}\_j)\$ is known as the kernel function. Kernel functions let the feature space be mapped into higher dimensional space without the need of calculating the transformation function (only the kernel is needed). This property makes them really powerful and easy to use. More details on SVM and kernel functions can be found in \cite{bishop2006pattern}

9) Page 4 line 25: please define all variables

We rewrote the section.

10) Page 5 line 3: maybe I missed it, but if you used LIBSVM or a derived tool please mention it, as this information could help other users

All the codes are written using the python scikit-learn package. We have explained it in the methodology section of the paper.

11) Page 5 line 25: I found this sentence somewhat disconnected.

We agree with you and have deleted the sentence.

12) Equation 3: define the class index.

**What does he mean?**

13) Page 5 line 29: H=0 usually means low entropy which could be seen as a pure prediction. Please clarify.

We rewrote the paragraph as follows:

where  $p_{i}\$  represents a set of probabilities that adds up to 1. H(x)=0 means that no new information was gained in the process of splitting, and H(x)=1 means that maximum amount of information was achieved. Ideally, the produced leaves will be pure and have low entropy H(x)=0 meaning all of the objects in the leaf are the same.

14) General comment: I would summarize for all methods the hyperparameters to be tuned. This is currently well done only for some of the methods.

We added a short subsection addressing the comment:

Machine learning methods are generally parametrized by a set of hyper-parameters \$\lambda\$. An optimal set of hyper-parameters \$\lambda\_{best}\$ will result in an optimal algorithm which minimizes the loss function that formally can be written as:

**\begin{equation}**

\lambda\_{best} = arg min L(X\_{test}; A(X\_{train}, \lambda))

**\end{equation}**

where A is the algorithm and \$X\_{test}\$ and \$X\_{train}\$ are test and training data. Searching to find the best set of hyper-parameters is mostly done by grid search method in which set of values on a predefined grid will be proposed. Implementing each of the proposed hyperparameters the algorithm will be trained, and the prediction results will be compared. Most algorithms have only few hyper-parameters. Depending on the learning algorithm, the size of training and test data sets the grid search can be a time consuming approach. Thus automatic hyper-parameter optimization has gain interest, details on the topic can be found else where \citep{feurer2019hyperparameter} }

15) Page 7 line 1: "a detailed description" this sentence and the introduction to various methods give the impression of ML as a closed list of techniques. As the Lidar community is not very

familiar to ML, I would mention that a vast number of other techniques exist. I would also mention ANNs and explain why those were not used in the present paper

The "a detailed description" refers to a Hastie et al., 2009's detailed description of all unsupervised learning methods and not to our explanation.

We added a paragraph in page 4, before explaining the ML algorithms which we used in this study:

\textcolor{blue}{Many algorithms have been developed for both supervised and unsupervised learning. In the following section, we introduce Support Vector Machine (SVM), Decision Trees, Random Forests and Gradient Boosting Tree Methods as part of ML algorithms that we have tested for sorting lidar profiles. We also describe The t-distributed Stochastic Neighbour Embedding Method and Density-based spatial clustering of applications with noise (DBSCAN) unsupervised algorithms which were used in this paper.}

\textcolor{blue}{Recently, Deep Neural Networks (DNNs) have received attention in the scientific community. In the Neural Network approach the loss function computes the error between the output scores and target values. The internal parameters (weights) in the algorithm are modified such that the error becomes smaller. The process of tuning the weights continues until the error is not decreasing anymore. A typical deep learning algorithm can have hundreds of millions of weights, input and target values. Thus the algorithm are really useful when dealing with large sets of images and text data. Although, DNNs are power full tools, they are acting as black boxes and important questions such as what features in the input data are more important will stay unknown. For the purpose of this study we decided to use the classical machine learning algorithms.}

16) I would remove Figure 2 (too much detail in comparison with the rest of the chapter), but this is a matter of personal taste.

We prefer to have this figure in the article.

17) Page 8 line 15: from and not form

Thanks for pointing out this typo.

18) Section 3.1: see general comment above, how was it ensured that enough separation between training situations and test situations is achieved (scans taken the same day/hour might not always achieve this). This point needs to be discussed carefully

As we responded to the first comment, our training and test data-sets contain scans from different years, months, and hours to make sure that we can achieve the generalization.

19) Section 3.2: In my opinion the discussion on TP/FP/TN/FN does not belong to the results but to the methods.

As we wanted to show the confusion matrix under the result and we found out putting the definitions of TP/FP/TN/FN in the methodology will require us to insert another short subsection. So, we decided to just squeeze the definition in the result.

20) Page 11 line 6: to estimate

Fixed

21) Page 14 line 11: anomalies and not anomolies

**Fixed**

22) General comment: why not making use of thresholds in order to achieve finer objectives, eg. "never tagging good scans as bad"? For example, with SVMs one can use the distance to the hyperplane to select events with a good likelihood of correct classification.

As the other two algorithms did better in the test and evaluation phase we did not attempt to use the SVM as our primary approach of classification. However, we definitely agree with you that we can use the distance to hyperplane to achieve a probability approach rather than a solid binary classification.

**Response to Reviewer 2:**

Thanks for the corrections and comments, we implemented all of them in our final version of the paper. We also went through the text carefully and proofread the article.

1. Why the authors use the term *scan* to indicate a lidar profile? I am used to refer to a scan strictly as for a scanning device, e.g., a scanning wind lidar will give a profile of wind strength and direction. The recording in time of the photon- counting signal when transformed into altitude is better called a profile.

We agree with you and we changed the term scan to profile throughout the paper.

2. When the authors use the relative pronoun "which", this shall come after comma otherwise "that" has to be used instead.

**Thanks for the comment. We corrected the paper accordingly.**

3. Plenty of punctuation, articles and auxiliary verbs are missing through the text. I have tried to highlight a part of them, but the authors should perform a thorough re-reading of the manuscript and correct these typos and errors.

Sect.1, Pg 2, In 20: the PCL acronym has already been defined. *done*

Sect.2, Pg 3, In. 8: you can remove "which are as follows". done

Sect.2, Pg 3, In. 23: "minimizes" done

Figure 1, caption: I'd say the line color in the left panel is Cyan rather than blue. Sect. 2.2, Pg 4, In 5: "we have tested" (remove "been") *done*

Sect. 2.4, pg6, In 3: As it is mentioned here for the first time, it could be useful to add the extended name of bagging in brackets "bagging (bootstrap aggregating)" . *done*

Sect. 2.5, pg 6, ln 28-29: the definition of overfitting should be provided in Sect. 2.4 when it is first introduced. *done*

Sect. 3.1, pg 9, In 4: ". . ..different years AND represent different. . ." done

Sect. 3.1, pg 9, In 21: change "True negatives (FN)" to "False negative (FN)" done

Sect. 3.1, pg 9, last paragraph: first you define precision and recall and then you present results of accuracy in Table 1. I would show Table 2 before Table 1, right below equations 9.

We changed the structure so that we presented the accuracy score and then we defined precision and recall.

Sect. 3.1, pg 10, In 5: "clear scans" done

Sect. 3.3, pg 13, ln 9: replace "smoke does not present" with "smoke is not present". Sec. 3.3., pg 13, ln 12: replace "during late June and early June 2002" with "during early June and late June 2002". *done*

Sec. 3.3., pg 13, In 13: "As, the smoke" without a comma *done*

Sec. 3.3., pg 13, In 16: "1961 lidar scans" done

Sect. 3.3, pg 14, ln 2: replace "To investigate if scans with layers are clustered to- gether the particle extinction . . . " with "To investigate if scans with layers are clustered together, the particle, extinction. . .".

Sect. 3.3, pg 14, In 10-11: please consider rephrasing and correcting the English. Sect. 3.3, pg 14, In 11: "no anomalies WERE detected" *done*

"We are planning to expand our unsupervised learning method to both Rayleigh and Nitrogen channels to be able to correctly identify and distinguish cirrus clouds from smoke traces in the UTLS.". Does the PCL have a depolarization chan- nel, or is in the forthcoming plans to implement one to discriminate between different particles based on their asphericity?

The PCL does not have a depolarization channel, which is the best way to distinguish smoke particles from ice. However, we have shown (Gamage et al, 2019) that is possible in our processing algorithms to use the multiple color measurements of the lidar to estimate the lidar ratio, which allows ice and smoke particles to be distinguished.

References:

Mahagammulla Gamage, S., Sica, R. J., Martucci, G., and Haefele, A.: Retrieval of temperature from a multiple channel pure rotational Raman backscatter lidar using an optimal estimation method, Atmos. Meas. Tech., 12, 5801–5816, https://doi.org/10.5194/amt-12-5801-2019, 2019.

[revised manuscript text omitted]

$$L(\boldsymbol{w}, b, \alpha) = \frac{1}{2} \left\| \boldsymbol{w}^2 \right\| - \sum_i \alpha_i \left( y_i(\boldsymbol{w}^{\mathsf{T}} \boldsymbol{x}_i + b) - 1 \right)$$
(3)

where  $\alpha_i$  are Lagrangian multipliers. Setting the derivatives of  $L(w, b, \alpha)$  with respect to w and b to zero:

$$\boldsymbol{w} = \sum_{i} \alpha_{i} y_{i} \boldsymbol{x}_{i} \tag{4}$$

$$\qquad \sum_{i} \alpha_{i} y_{i} = 0. \tag{5}$$

Thus we can rewrite the Lagrangian as:

$$L(\boldsymbol{w}, b, \alpha) = \sum_{i} \alpha_{i} - \frac{1}{2} \sum_{i} \sum_{j} \alpha_{i} \alpha_{j} y_{i} y_{j} \boldsymbol{x}_{i}^{\mathsf{T}} \boldsymbol{x}_{j}$$
(6)

It is clear that the optimization process only depends on the dot product of the samples.

Many real world problems involve nonlinear data sets in which the above methodology will fail. To tackle the non-linearity,
using a non-linear function Φ(x) the feature space is mapped into higher dimensional feature space. The Lagrangian function can be re-written as:

$$L(\boldsymbol{w}, b, \alpha) = \sum_{i} \alpha_{i} - \frac{1}{2} \sum_{i} \sum_{j} \alpha_{i} \alpha_{j} y_{i} y_{j} k(\boldsymbol{x}_{i}, \boldsymbol{x}_{j})$$

$$\tag{7}$$

$$k(\boldsymbol{x}_i, \boldsymbol{x}_j) = \Phi(\boldsymbol{x}_i)^{\mathsf{T}} \Phi(\boldsymbol{x}_j) \tag{8}$$

where  $k(x_i, x_j)$  is known as the kernel function. Kernel functions let the feature space be mapped into higher dimensional 150 space without the need of calculating the transformation function (only the kernel is needed). More details on SVM and kernel functions can be found in Bishop (2006)

To use SVM as a multi-class classifier, some adjustments need to be made to the simple SVM binary model. Methods like a directed acyclic graph, one-against-all, and one-against-others are among the most successful techniques for multi-class classification. Details about these methods can be found in Knerr et al. (1990).

**155 2.3 Decision Trees Algorithms**

Decision trees are nonparametric algorithms that allow complex relations between inputs and outputs, to be modeled. Moreover, they are the foundation of both random forest and boosting methods. A comprehensive introduction to the topic can be found in Quinlan (1986), here, we briefly describe how a decision tree is built.

- A decision tree is a set of (binary) decisions represented by an acyclic graph directed outward from a root node to each leaf.
  160 Each node has one parent (except the root), and can have two children. A node with no children is called a leaf. Decision trees can be complex depending on the data set. A tree can be simplified by pruning, which means leaves from the upper parts of the trees will be cut. To grow a decision tree, the following steps are taken.
  - Defining a set of candidate splits: We should answer a question about the value of a selected input feature to split the data set into two groups.
- Evaluating the splits. Using a score measure, at each node, we can decide what the best question is to be asked and what the best feature is to be used. As the goal of splitting is to find the purest learning subset that is in each leaf, we want the output labels to be the same; called purifying. Shannon Entropy (see below) is used to evaluate the purity of each subgroup. Thus, a split that reduces the entropy from one node to its descendent is favorable
  - Deciding to stop splitting. We set rules to define when the splitting should be stopped, and a node becomes a leaf. This decision can be data-driven. For example, we can stop splitting when all objects in a node have the same label (pure

170

node). The decision can be defined by a user as well. For example, we can limit the maximum depth of the tree (length of the path between root and a leaf).

In a decision tree, by performing a full scan of attribute space the optimal split (at each local node) is selected, and irrelevant attributes are discarded. This method allows us to identify the attributes that are most important in our decision-making process.

175 In summary, the simplicity of decision trees makes them suitable algorithms for both classification and regression processes.

The metric used to judge the quality of the tree splitting is Shannon entropy (Shannon, 1948). Shannon Entropy describes the amount of information gained with each event and is calculated as follows:

$$H(x) = -\Sigma p_i \log p_i \tag{9}$$

where  $p_i$  represents a set of probabilities that adds up to 1. H(x) = 0 means that no new information was gained in the process 180 of splitting, and H(x) = 1 means that maximum amount of information was achieved. Ideally, the produced leaves will be pure and have low entropy (meaning all of the objects in the leaf are the same).

**2.4 Random Forests**

The Random forest (RF) method is based on "growing" an ensemble of decision trees that vote for the most popular class. Typically the bagging (bootstrap aggregating) method is used to generate the ensemble of trees (Breiman, 2002). In bagging,

- to grow the  $k_{th}$  tree, a random vector  $\theta_k$  from the training set is selected. The  $\theta_k$  vector is independent of the past vectors  $(\theta_1, ..., \theta_{k-1})$  but has the same distribution. Then, by selecting random features, the  $k_{th}$  tree is generated. Each tree is a classifier  $(h(\theta_k, \mathbf{x}))$  that casts a vote. During the construction of decision trees, in each interior node, the Gini index is used to evaluate the subset of selected features. The Gini index is the measure of impurity of data (Lerman and Yitzhaki, 1984; Liaw et al., 2002). Thus, it is desired to select a feature that results in a greater decrease in the Gini index (partitioning the data into
- 190 distinct classes). For a split at node n the index can be calculated as:  $1 \sum_{i=1}^{2} P_i^2$  where  $P_i$  is the frequency of class *i* in the node *n*. Finally, the class label is determined via majority voting among all the trees (Liaw et al., 2002).

One major problem in ML is when the algorithm becomes too complicated and perfectly fits the training data points, it looses its generality and performs poorly on the testing set. This problem is known as overfitting. For RF, increasing the number of trees can help with the overfitting problem. Other parameter that can significantly influence RFs is the tree depth, growing more

195 trees in a forest yield a smaller prediction error. Finding the optimal depth of each tree is a critical parameter. While leaves in a short tree may contain heterogeneous data (the leaves are not pure), tall trees can suffer from poor generalization (overfitting problem). Thus, the optimal depth provides a tree with pure leaves and great generalization. Detailed discussion on the RFs can be found in Liaw et al. (2002).

**2.5 Gradient Boosting Tree Methods**

200 Boosting methods are based on the idea that combining many "weak" approximation models (a learning algorithm that is slightly more accurate than 50%) will eventually boost the predictive performance (Knerr et al., 1990; Schapire, 1990). Thus, many "local rules" are combined to produce highly accurate models.

In the gradient boosting method, simple parametrized models (base models) are sequentially fitted to current residuals (known as pseudo-residuals) at each iteration. The residuals are the gradients of the loss function (they show the difference

205 between the predicted value and the true value) that we are trying to minimize. The Gradient Boosting Trees (GBT) algorithm is a sequence of simple trees generated such that each successive tree is grown based on the prediction residual of the preceding tree with the goal of reducing the new residual. This "additive weighted expansion" of trees will eventually become a strong classifier (Knerr et al., 1990). This method can be successfully used even when the relation between the instances and output values are complex. Compared to the RF model, which is based on building many independent models and combining them 210 (using some averaging techniques), the gradient boosting method is based on building sequential models.

One major problem in ML is when the algorithm becomes too complicated and perfectly fits through the training data points. However, the algorithm looses its generality and performs poorly in the testing set. This is known as overfitting. 
[revised manuscript text omitted]

| Nitrogen channel |           |        |       |            |        |  |       |            |        |  |  |
|------------------|-----------|--------|-------|------------|--------|--|-------|------------|--------|--|--|
| Scon             | Drecision | Pacall |       | LR channel |        |  |       | HK channel |        |  |  |
| Tupe             | vpe       |        | Scan  | Precision  | Recall |  | Scan  | Precision  | Recall |  |  |
| Туре             |           |        | Type  |            |        |  | Type  |            |        |  |  |
| Cloud            | 0.94      | 0.91   | Type  |            |        |  | 1) PC |            |        |  |  |
| Clear            | 0.96      | 0.98   | Clear | 0.99       | 0.99   |  | Clear | 0.98       | 1.00   |  |  |
| Dad              | 1.00      | 1.00   | Bad   | 0.96       | 0.95   |  | Bad   | 0.98       | 0.94   |  |  |
| Dau              | 1.00      | 1.00   |       |            |        |  |       |            |        |  |  |

Table 2. Precision and recall values for the nitrogen, LR, and HR channels. The precision and recall values are calculated using the GBT model.

- 305 accuracy results on HR and LR channels. The accuracy of the model on the training set on the LR channel for both RF and GBT are 99% and on the test set are 98%. The precision and recall values for the clears profiles are close to unity and for the bad profiles they are 0.95 and 0.96 respectively. The HR channel also has a high accuracy of 99% in the training set for both RF and GBT, and the accuracy score in the testing set is 98%. The precision and recall values in Table 2 are also similar to the LR channel.
- 310 For the nitrogen channel the GBT algorithm has the highest accuracy of 95% while the RF algorithm with accuracy of 94%. The confusion matrix of the test result for the GBT algorithm (the one with the highest accuracy) is shown in Fig. 3 (left panel). The algorithm can perform almost perfectly on distinguishing bad profiles (only one bad scan was wrongly labeled as cloudy). The cloud and clear profiles for most profiles are labeled correctly; however for few profiles the model mislabeled clouds for clear profiles.

Figure 3. The confusion matrices for nitrogen channel (left), LR channel (middle), and HR channel (right). In a perfect model, the offdiagonal elements of the confusion matrix are zeros.

**315 3.2 Unsupervised ML Results**

The t-SNE algorithm clusters the measurements by means of pairwise similarity. The clustering can differ from night to night due to atmospheric and systematic variability. On nights where most profiles are similar, fewer clusters are seen, and on other nights when the atmospheric or the instrument conditions are more variable, more clusters are generated. The t-SNE makes

clusters but does not estimate how many clusters are built, and the user must then estimate the number of clusters. To automate

this procedure, after applying the t-SNE to lidar measurements we use the DBSCAN algorithm to estimate the number of 320 clusters. The second step of applying DBSCAN is used to estimate the number of generated clusters by t-SNE.

To demonstrate how clustering works, we show measurements from the PCL LR channel and the nitrogen channels. Here, using the t-SNE on May 15, 2012 that contains both bad and good profiles. We also show the clustering result for the nitrogen channel on May 26 2012. We chose this night because at the beginning of the measurements the sky was clear but the sky became cloudy.

On the night of May 15 2012, the t-SNE algorithm generates three distinct clusters for the LR channel (Fig. 4). These cluster correspond to different types of lidar return profiles. Figure 5 (left panel) shows all the signals for each of the clusters. The maximum number of photon counts and the value and the height of the background counts are the identifiers between different clusters. Thus, cluster 3 with low background counts and high maximum counts represents a group of profiles which

330

325

are labeled as good profiles in our supervised algorithms. Cluster 1 represents the profiles with lower than normal laser powers, and clusters 2 shows profiles with extremely low laser powers. To better understand the difference between these clusters, Fig. 5 (right panel) shows the average signal. Furthermore, the outliers of cluster 3 (shown in black) identify the profiles with extremely high background counts. This result is consistent with our supervised method, in which we had good profiles (here is cluster 3), and bad profiles which are profiles with lower laser power (here are cluster 1 and 2).